# Direct-Acting Antiviral Agents for Hepatitis C Virus Infection—From Drug Discovery to Successful Implementation in Clinical Practice

**DOI:** 10.3390/v14061325

**Published:** 2022-06-17

**Authors:** Christopher Dietz, Benjamin Maasoumy

**Affiliations:** Department of Gastroenterology, Hepatology and Endocrinology, Medical High School Hannover, 30625 Hannover, Germany; dietz.christopher@mh-hannover.de

**Keywords:** HCV, hepatitis C virus, DAA, direct-acting antiviral agents, treatment, antiviral

## Abstract

Today, hepatitis C virus infection affects up to 1.5 million people per year and is responsible for 29 thousand deaths per year. In the 1970s, the clinical observation of unclear, transfusion-related cases of hepatitis ignited scientific curiosity, and after years of intensive, basic research, the hepatitis C virus was discovered and described as the causative agent for these cases of unclear hepatitis in 1989. Even before the description of the hepatitis C virus, clinicians had started treating infected individuals with interferon. However, intense side effects and limited antiviral efficacy have been major challenges, shaping the aim for the development of more suitable and specific treatments. Before direct-acting antiviral agents could be developed, a detailed understanding of viral properties was necessary. In the years after the discovery of the new virus, several research groups had been working on the hepatitis C virus biology and finally revealed the replication cycle. This knowledge was the basis for the later development of specific antiviral drugs referred to as direct-acting antiviral agents. In 2011, roughly 22 years after the discovery of the hepatitis C virus, the first two drugs became available and paved the way for a revolution in hepatitis C therapy. Today, the treatment of chronic hepatitis C virus infection does not rely on interferon anymore, and the treatment response rate is above 90% in most cases, including those with unsuccessful pretreatments. Regardless of the clinical and scientific success story, some challenges remain until the HCV elimination goals announced by the World Health Organization are met.

## 1. Introduction

The history of hepatitis C research is a success story where the interplay of applied clinical research and basic science led to the discovery of a new virus and the establishment of a curative treatment within little more than twenty years (Figure 1). The tremendous impact of hepatitis C virus (HCV) research became even clearer when HCV researchers were awarded the 2016 Lasker award and the 2020 Nobel Prize in Medicine [1]. The scientific advances in HCV research were based on a “bedside-to-bench-to-bedside” approach that can serve as a role model for other fields of biomedical research. In the following article, we review the important steps of the HCV discovery and the characterization of viral properties that finally led to the development of direct-acting antiviral agents as cures for HCV infection.

### 1.1. Discovery of the Hepatitis C Virus

In the 1970s, transfusion-associated cases of hepatitis were frequently reported, resulting in the descriptive term of post-transfusion hepatitis. Soon, it was found that known hepatotropic viruses such as hepatitis A and B viruses were not responsible for the new type of hepatitis. Consequently, it was named non-A-non-B hepatitis (NANBH) [3]. After this clinical observation of NANBH, years of extensive basic research followed. In 1989, two papers were published in the same *Science* edition that described the isolation of viral cDNA clones [4] that allowed for the identification of NANBH antibodies, finally resulting in the discovery of HCV [5].

### 1.2. Epidemiology and Natural History of Hepatitis C Virus Infection

Acute hepatitis develops within four to twelve weeks after HCV infection [6]. A rise in bilirubin leading to clinically visible jaundice is possible. However, the symptoms are often unspecific, and asymptomatic disease courses are common. Consequently, individuals can be infected unknowingly. In up to 80% of cases, acute hepatitis C develops into chronic HCV infection [7]. Chronic hepatitis with ongoing hepatic damage bears the risk of liver cirrhosis. In a large meta-analysis, liver cirrhosis was present in 16% of individuals after 20 years of chronic HCV infection. Risk factors for the development of liver cirrhosis include the presence of chronic hepatitis with elevated ALT levels, large amounts of alcohol intake, or coinfection with the hepatitis B virus [7]. In addition, chronic HCV infection is a major risk factor for the development of hepatocellular carcinoma (HCC). Typically, HCC develops in cirrhotic liver tissue. Estimates of annual HCC incidence range from 0.5% to 10% [8]. In addition to hepatic involvement, HCV infection can also be linked to extrahepatic manifestations, as summarized in Figure 2. However, a causal relationship has not been demonstrated for all the extrahepatic symptoms that have been described in the past [9]. HCV is transmitted via parenteral pathways. In fact, contaminated blood products are a major source of infection, which led to the initial descriptive term of post-transfusion hepatitis. Today, with improved safety measures in transfusion medicine, infections are caused by intravenous drug abuse and the associated needle sharing [10]. Medical staff is another risk group, as injuries with contaminated instruments such as canula can transmit HCV [11].

Worldwide, 1.5 million new HCV infections and 29,000 HCV-related deaths per year are assumed by the World Health Organization (WHO) [12]. The substantial health burden caused by HCV infection and viral hepatitis in general prompted the formulation of the hepatitis extinction goals by the WHO in 2016. In detail, the WHO pursues the ambitious goal of an 80% reduction in new HCV infections by the year 2030 [13].

### 1.3. Discovery of the HCV Replication Cycle

After the clinical description of NANBH and the discovery of its causative agent, HCV viral properties and the HCV replication cycle were subjects of intensive research. It was found that HCV is a positive single-stranded RNA virus that likely belongs to the family of Flaviviridae [4]. After internalization of the virus, the ssRNA(+) is released into the cytoplasm where host ribosomes on the endoplasmic reticulum translate the genetic material into the HCV polyprotein [14]. The viral proteins can be divided into structural and nonstructural (NS) proteins. Nonstructural proteins are needed for viral amplification and were seen as potential drug targets [14].

Even though HCV had been discovered, a cell culture was not available for the coming years. Consequently, cDNA clones generated from viral RNA extracted from the sera of infected individuals served as models for detailed analysis of the genome structure. It was suggested that an HCV polyprotein containing approximately 3000 amino acids is translated from a single open reading frame [15]. Comparisons to the genetic features of other viruses soon made it clear that HCV is related to pestiviruses, as well as flaviviruses, both genera of the family of Flaviviridae. HCV was deemed an unusual virus. However, the relative proximity to flaviviruses suggested that the HCV genome might be organized in a similar way. Flaviviridae are characterized by hydrophobic polyproteins that are cleaved by viral and host proteases into structural and nonstructural proteins [16]. In analogy, an HCV polyprotein was assumed. Computer-based approaches suggested potential cleavage sites, further promoting the understanding of potential structural and nonstructural proteins in HCV [15,16]. In a next step, in vitro protein expression assays allowed for a deeper insight into the actual structure of viral proteins. Therefore, expression plasmids were generated from HCV cDNA clones. These plasmids encoded 980 amino-terminal amino acid residues of the open reading frame. This work by Hijikata led to the identification of the two proteins, p22 and p19, as well as two glycoproteins named gp35 and gp70. It was suggested that the cleavage between the proteins depended on signal peptidases in the lumen of the endoplasmic reticulum, with the exception of gp70 and p19. Finally, gp35 and gp60 were characterized as major viral envelope proteins [17]. Similarly, additional cleavage products of HCV polyprotein, including the nonstructural proteins NS2, NS3, NS4A, NS4B, NS5A, and NS5B, were discovered (Table 1) [18]. The detailed description of viral proteins was complemented through research into the according molecular function. NS3 was identified as a viral protease catalyzing the cleavage between NS3 and NS4. This became evident in experiments where the replacement of a serine residue within the NS3 genome area blocked the cleavage between NS3/4 and NS4/5 [19]. Soon, it was found that, in addition to NS3, NS4A is also involved in the cleavage of the link between NS3/4A and NS5A/5B [20,21]. With viral proteases, a first potential therapeutic target was discovered (Figure 3). However, the exact mechanisms behind viral RNA synthesis were still hidden. In 1990, it was postulated that NS5B might be the responsible RNA polymerase [22]. In fact, six years later, Behrens et al. confirmed that NS5B was acting as an RNA-dependent RNA polymerase [23].

In parallel to intensive research on HCV polyprotein processing and viral replication, the key mechanisms of viral entry into hepatocytes were unraveled. First, CD81 was described as a ligand of the envelope protein E2, formerly known as gp70 [24]. HCV endocytosis was also found to be dependent on the LDL receptor [25]. Further steps of viral entry were attributed to the human scavenger receptor class B type 1 [26], as well as claudin-1 [27] and occludin [28].

The establishment of in vitro replication models was the next essential achievement, enabling the evaluation of antiviral agents and their potencies. Using RNA isolated from an HCV-infected individual, replicons were generated. These replicons were transfected into a human hepatoma cell line where autonomous replication was observed [29]. Another group was able to create replicons from different HCV genotypes that replicated in hepatoma cell lines. They also showed the suppression of viral replication by exposing cell cultures to interferon alpha, emphasizing the value of such replication models [30]. Finally, cell culture systems were successfully used to generate viral replicates that were infectious for chimpanzees [31].

## 2. Development of Direct-Acting Antiviral Agents

With working cell culture systems at hand, the quest for efficient therapeutic agents began (Figure 3). At this time, the research focus shifted back from bench to bedside, where the “non-A-non-B post-transfusion hepatitis” had been observed in the 1970s. Early clinical trials investigated the efficacy of protease inhibitors, as nonstructural proteins were seen as promising therapeutic targets. This led to the identification of BILN-2061, a small molecule inhibiting NS3. In fact, this inhibition led to significant declines in the viral loads of patients included in a phase I trial of BILN-2061 [32]. Subsequent clinical trials evaluated short-term response and observed a decline in viral load regardless of the stage of liver fibrosis [33]. However, when extending the selection criteria to HCV genotypes 2 and 3, a double-blind pilot study showed mixed responses. It was assumed that this observation was at least partially due to a decreased affinity of BILN-2061 to NS3 in these genotypes [34]. Regardless of BILN-2061’s ability to decrease viral load and its good safety profiles in the mentioned trials, the suspicion of cardiac toxicity in animal models led to the termination of further investigations [35]. However, in 2011 the first two protease inhibitors were approved for the treatment of hepatitis C. Telaprevir [36,37,38] and boceprevir [39,40] were evaluated in addition to the former standard treatment of PEG-interferon and ribavirin (Peg-IFN-RBV) in patients infected with genotype 1. In treatment-naïve patients, telaprevir led to a sustained virologic response (SVR) in 72% (*n* = 388/540). In up to 66% of the cases of patients with a history of unsuccessful Peg-IFN-RBV treatment, retreatment with additional telaprevir led to SVR [38]. In a similar way, this was shown for boceprevir with an SVR of up to 68% (*n* = 213/311) in treatment-naïve [39] and up to 66% in treatment-experienced individuals [40]. With the approval of the first two DAAs, a milestone in HCV research was reached. For the first time, an additional HCV-specific therapy could be added to the former standard therapy of Peg-IFN-RBV that also helped achieve SVR in those with initially unsuccessful treatment attempts [41,42]. However, the new treatment strategy still relied on the use of peg-IFN and ribavirin. Ever since the use of interferon, physicians have had to account for significant side effects, limiting the use of interferon to a selected patient population. For example, interferon cannot be used in patients with decompensated liver cirrhosis or psychiatric disorders [43]. Data from real-world experience with triple therapies, including one PI, suggested that a low platelet count as surrogate for advanced liver disease was associated with a higher risk of treatment failure and side effects [43].

At this time, the genetic variability of HCV was challenging. Following the first cell culture models based on genotype 1b [29], early DAA were optimized for this genotype. As a result, the barrier of resistance and SVR rates tended to be higher in genotype 1b compared to 1a patients, e.g., for some PI-based regimens. Moreover, many of the initially developed PI had no or only minor efficacy in genotype 2 or 3 infections [45]. Genotypes 1 and 3 are the most frequent genotypes worldwide [46], and for genotype 3, a higher risk of liver cirrhosis and HCC was reported in a US cohort [47]. Over time, the genetic variability of HCV was further explored until, in 2015, genotype 7 was discovered [48]. Consequently, the development of pangenotypic treatment regimens was an important next step.

In addition to the concept of protease inhibition, the characterization of HCV polyprotein and nonstructural viral proteins allowed for the development of additional drug classes. It was shown that the inhibition of NS5A leads to a decline in viral replication [49]. Chemical genetics were used to characterize promising agents interfering with NS5A [50]. In a proof-of-concept study, SVR was achieved with a combination of the NS5A inhibitor daclatasvir and the protease inhibitor asunaprevir. These results prepared the ground for IFN-free treatment regimens. A phase III trial investigating the safety and efficacy of daclatasvir and asunaprevir in individuals with genotype 1b infections showed SVR rates of 90% in treatment-naïve patients and 82% in treatment-experienced and IFN-ineligible patients [51]. Another therapeutic target of interest was the viral polymerase NS5B [52]. Under the name sofosbuvir, the first NS5B inhibitor became available. Sofosbuvir combined with ribavirin with and without interferon was evaluated in numerous clinical trials, including patients with genotypes 2 and 3 (see [53] for summary). In the ELECTRON trial, it was shown that 12 weeks of SOF and RBV without IFN was sufficient to reach high rates of SVR in individuals infected with genotypes 2 and 3. Furthermore, the pangenotypic efficacy was an outstanding advantage of SOF [54]. A relatively low risk for drug–drug interactions simplified its integration into clinical practice [55], which was a benefit especially for patients coinfected with HIV and intake of antiretroviral therapy [56,57]. As of today, fixed-dose DAAs with treatment durations of eight or twelve weeks represent the established standard treatment for HCV infection. In 2020, the European Association for the Study of Liver disease (EASL) published their final update of the HCV guidelines, highlighting the success of extensive research into DAAs. The current treatment recommendations mainly involve four different combinations of DAAs, which are presented in the following lines.

### 2.1. Sofosbuvir/Velpatasvir (SOF/VEL)

The combination of sofosbuvir as an NS5B inhibitor and velpatasvir as an NS5A inhibitor represents a PI-free treatment regimen. The combination of SOF/VEL is advantageous in patients with decompensated liver cirrhosis, as PIs are contraindicated in this patient population [58]. Strong P-gp and CYP-inducers should be avoided due to a risk of loss of antiviral efficacy [59,60]. The antiviral efficacy of SOF/VEL was proved in the ASTRAL trial series. The placebo-controlled ASTRAL-1 trial evaluated the efficacy against all common HCV genotypes, with the exception of genotype 3. A 95% confidence interval for SVR rates between 98% and >99% indicated excellent antiviral activity [61]. Patients infected with genotypes 2 or 3 were included in the ASTRAL-2 and 3 trials [62]. The combination of SOF/VEL was superior to SOF/RBV, and after 12 weeks of treatment, SVR rates between 95% (genotype 3) and 99% (genotype 2) were reached [62]. As a PI-free regimen, SOF/VEL was also evaluated in patients with Child–Pugh B liver cirrhosis. The ASTRAL-4 trial compared SOF/VEL with SOF/VEL and RBV. The highest SVR rates were reached in the treatment group receiving the combination of SOF/VEL and RBV for a duration of 12 weeks. In this vulnerable patient population with an urgent need for treatment, remarkable SVR rates of 94% were reached [63]. In the ASTRAL-5 trial, it was finally shown that SOF/VEL was also safe and effective in patients coinfected with HIV-1 [64].

### 2.2. Glecaprevir/Pibrentasvir (G/P)

The PI glecaprevir and the NS5A inhibitor pibrentasvir form another fixed-dose pangenotypic DAA combination recommended by the current guidelines [58]. The inclusion of a PI limits the use of G/P in patients with compensated liver disease. Shorter treatment durations are possible with G/P. In a large, integrated analysis including a total of 2041 noncirrhotic patients, SVR rates reached 98% after eight weeks [65]. An Italian study group provided real-world data from 723 patients, of which 89% were treated with G/P for eight weeks while, in the rest, longer treatment durations were tested. Comparable to the results from clinical trials, SVR rates were 94% in the intention-to-treat analysis. In addition, good tolerability and drug safety were reported [66]. Similarly, this was shown in a large, German, real-world cohort [67], making G/P an advantageous treatment option in those patients where shorter treatment durations might improve treatment adherence.

### 2.3. Grazoprevir/Elbasvir

The combination of the PI grazoprevir and the NS5A inhibitor elbasvir is another PI-containing fixed-dose DAA regimen. Similar to G/P, grazoprevir/elbasvir should not be used in patients with decompensated liver cirrhosis and is limited to patients infected with genotype 1b. An integrated retrospective analysis of several clinical trials revealed good viral responses in individuals infected with genotype 1b, with an SVR rate of 97% [68]. Previously, it had been shown that the risk of therapeutic failure was higher in cases of infections with genotype 1a. However, mutation analyses scanning the viral genome for resistance-associated substitutions in the NS5A gene and adopting treatment durations helped increase the SVR rate in genotype 1a [69].

### 2.4. Sofosbuvir/Velpatasvir/Voxilaprevir (SOF/VEL/VOX)

Even though SVR rates are generally high in the era of DAA treatment, therapeutic failure occurs. The POLARIS trials evaluated the efficacy of SOF/VEL complemented by the PI voxilaprevir in patients with previous unsuccessful DAA treatment [70]. In POLARIS-1, the majority of patients had either been treated with an NS5A inhibitor and an NS5B inhibitor or with an NS5A inihibtor and an NS3 inhibitor. Patients pretreated with NS5A inhibitors were excluded from the POLARIS-4 trial. In both studies, the triple DAA was given for 12 weeks. SVR rates between 96% (POLARIS-1) and 98% (POLARIS-4) were observed, and good pangenotypic efficacy was documented. These promising results were further followed up in real-world settings, confirming SOF/VEL/VOX as a treatment option in DAA-experienced patients [71,72].

### 2.5. Current Treatment Strategies

The availability of pangenotypic, fixed-dose DAA regimens has simplified treatment recommendations. As of today, HCV therapy can be initiated even without the need for genotyping, improving the access to therapies. Following a simplified scheme, DAA-unexperienced patients without liver cirrhosis can be treated with twelve weeks of SOF/VEL or eight weeks of G/P [58]. In cases of compensated liver cirrhosis and a history of non-DAA therapy, the treatment duration with G/P should be extended to twelve weeks, while no adjustment is needed for SOF/VEL. If the genotype is available, treatment decisions can be modified accordingly. One example is the use of grazoprevir/elbasvir, which should only be administered in patients with genotype 1b [58]. Patients with decompensated liver cirrhosis belong to one of the most vulnerable subgroups. The benefit of DAA treatment in this cohort has been confirmed with real-world analyses. In one study including 409 patients with decompensated liver cirrhosis, successful DAA treatment was associated with an improvement in the MELD score. Patients with low sodium, ages ≥ 65, and low albumin (<35 d/L) were the least likely to benefit [73]. In decompensated liver cirrhosis, liver transplantation is a potential cure. Therefore, the effects of DAA in patients on transplant waiting lists has been discussed. Indeed, Belli et al. demonstrated that improvements in liver function with a consecutive decline in MELD scores led to delisting in a substantial amount of patients (33.3% at 60 weeks after DAA initiation), especially in those with a low MELD scores [74]. Ultimately, it was shown that the number of patients with HCV-related liver disease on a liver transplant waiting list in Spain was lowered by 2018 [75]. This emphasizes the clinical impact DAAs developed within years after their implementation into clinical practice. However, in order to develop an even more profound clinical impact at a population-wide level, infected individuals have to be detected more frequently and, consequently, referred to therapy. Large screening campaigns can, therefore, be considered as an important step towards HCV elimination, as aspired by the WHO and formulated in the previously mentioned HCV extinction goals [13]. A remarkable screening project on a large scale was recently conducted in Egypt. With the ambitious goal of screening every adult for HCV infection, a country-wide screening campaign orchestrated by the Egyptian Ministry of Health was set up. A total of 50 million people were screened within 7 months. HCV-viremia was detected in 1.1 million citizens, of which 92% started DAA treatment [76].

### 2.6. Developing an HCV Vaccination—A Remaining Challenge

While further screening campaigns with easy access to therapy are important, the development of an HCV vaccine is a remaining challenge. For hepatitis B, it was impressively shown that the availability of a working vaccine in combination with vaccination programs could lead to a dramatic decline in HBV-related liver disease and HBsAg seroprevalence [77,78]. In contrast to hepatitis B, there is currently no vaccine available against HCV. One major challenge in vaccine development is the genetic variability of HCV within one host, leading to quasispecies and inhibiting effective neutralization by the secreted antibodies [79]. Recently, a large phase II trial including 548 individuals with high risk for HCV infection evaluated the efficacy of a recombinant chimpanzee adenovirus 3 vector vaccine with a subsequent booster using a recombinant-modified vaccinia-virus. Even though T-cell responses against the encoded nonstructural proteins were measured, the development of chronic HCV infections was not prevented [80]. In addition to the nonstructural proteins used as immunologic targets, other approaches have focused on viral envelope proteins in order to prevent chronic HCV infection [80,81].

## 3. Conclusions

The development of DAAs as cures for HCV infection was a milestone in biomedical research. The fruitful interplay of applied clinical science and basic research was remarkable. The initial observation of unclear cases of post-transfusion hepatitis at the bedside of patients stimulated scientific curiosity and led to the description of HCV as the causative agent. The in-depth characterization of viral properties resulted in the discovery of potential therapeutic targets. While the quest for efficient, direct-acting antiviral agents had begun, clinicians were still using interferon with limited antiviral efficacy to protect patients from HCV-related chronic liver disease. Finally, the first DAA-containing treatment regimen was approved in 2011. However, IFN was still part of that treatment regimen, and clinicians, as well as scientists, were aiming for IFN-free HCV treatments. In 2013, the approval of SOF finally marked the beginning of an IFN-free era with excellent antiviral efficacy and SVR rates above 90%, even in those with unsuccessful IFN pretreatments. As of today, the remaining challenges include the detection of HCV-infected individuals and the referral to treatment, especially in low-income countries and communities with difficult access to health care. Ultimately, the development of a working HCV vaccine can help achieve the WHO HCV extinction goals in the coming years.

## Figures and Tables

**Figure 1 viruses-14-01325-f001:**
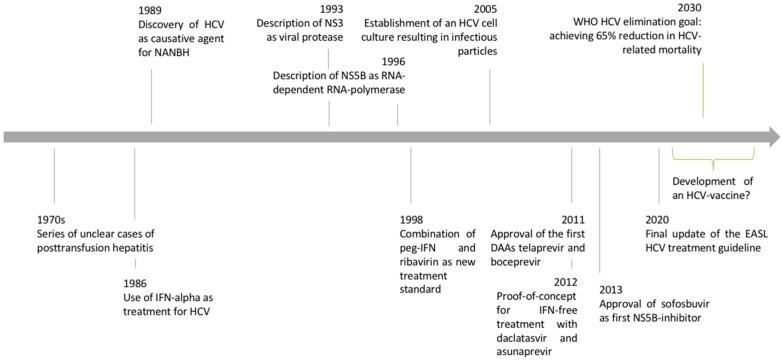
Timeline of HCV discovery and advances in clinical and basic virologic research. Adapted from [2].

**Figure 2 viruses-14-01325-f002:**
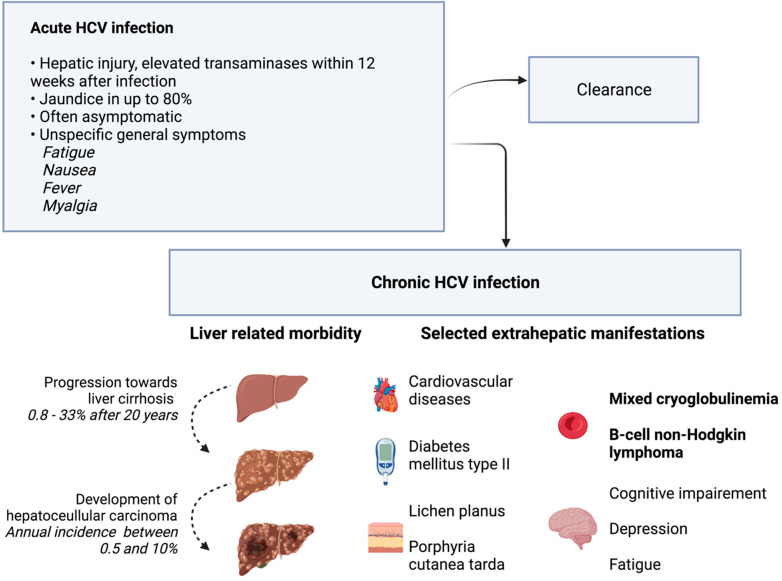
Clinical presentation and natural course of HCV infection. In addition to the acute and chronic clinical picture, selected extrahepatic manifestations are shown. Phenomena with the strongest evidence for association with chronic HCV infection are printed in bold font. The other phenomena were observed with higher prevalence in HCV-infected individuals than in controls, but evidence was less strong [9]. Additional references [7,8]. Created with BioRender.com.

**Figure 3 viruses-14-01325-f003:**
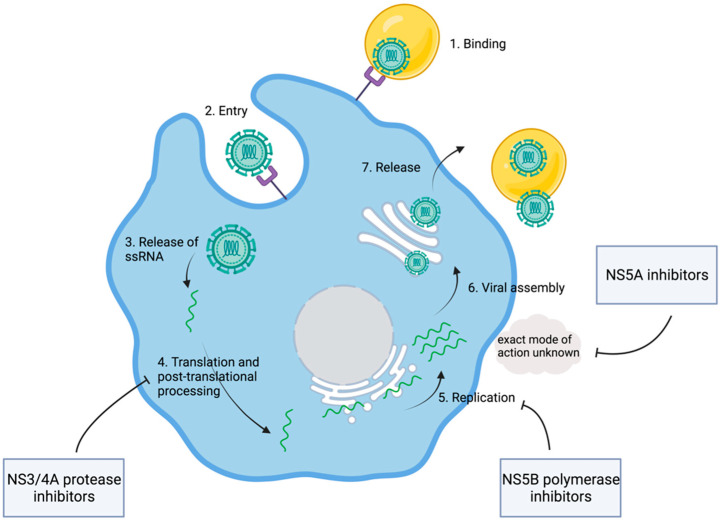
Viral replication and points of attack of direct-acting antiviral agents. Shown are key steps of viral replication and the modes of action of different DAAs. After binding and cellular entry of the viral particle, the ssRNA is released and translated into the HCV polyprotein. DAAs interfere with the viral replication at different stages. NS3/4A protease inhibitors block the processing of the HCV polyprotein. NS5B inhibitors interfere with the viral RNA polymerase. For NS5A inhibitors, an interaction with viral and host proteins is assumed, while the exact mode of action is unknown. Adapted from [44]. Created with BioRender.com.

**Table 1 viruses-14-01325-t001:** Nonstructural proteins of the hepatitis C virus with according molecular functions and targeting by currently recommended direct-acting antiviral drugs.

Nonstructural Proteins	Molecular Function	Drug Targeting
NS3	Viral protease cleavage between NS3/NS4 and NS4/NS5	Protease inhibitors, e.g., glecaprevir, voxilaprevir, grazoprevir
NS4A	Cofactor in viral proteolytic activity of NS3	/
NS5A	Mediates interactions between viral and host proteins	NS5A inhibitors, e.g., velpatasvir, pibrentasvir, elbasvir
NS5B	RNA-dependent RNA polymerase	NS5B inhibitor, e.g., sofosbuvir

## Data Availability

Not applicable.

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
