# Peer review of "Direct-Acting Antiviral Agents for Hepatitis C Virus Infection—From Drug Discovery to Successful Implementation in Clinical Practice"

_viruses, 2022, doi:10.3390/v14061325_

Round 1
Reviewer 1 Report
This is a very well written review of direct acting antivirals for HCV that are now in the clinic treatment during the early discovery of their molecular targets. Rarely (maybe never before) have I submitted a review without making some suggestion for change; however, in this case I wont.
The only negative I noticed was a format/layout error of Line 69 page 1 that separated the Title from the actual Table 1 on page two. I expect that will be caught/handled in the final layout.
This reviewer appreciates how complicated the development of DAA for HCV actually was and that the authors navigated the subject skillfully from a neutral perspective. While this reviewer would like to have seen more discussion of the basic science involved, doing so would certainly change the current readability.
As written, it covers major milestones and provides excellent, referencing for those wanting to dig deeper.
I suggest accepted as submitted.
Author Response
Please see attachement

Reviewer 2 Report
Dear Authors,
The paper entitled “Direct acting antiviral agents for Hepatitis C Virus infection– from drug discovery to successful implementation in clinical practice” has has merit and will probably give a significant contribution to the field. However, some improvements should be made to make it suitable for publication.
- The study reviews many important points, nevertheless, the HCV genetic variability was not addressed. Genotyping was crucial for achieving SVR during the treatment of chronic carriers (before the emergence of pan-genotypic drugs). Moreover, HCV quasisepecies are a major obstacle to vaccine development.
-Table 1 is not a table properly. Clinical presentation of HCV infection should be shown in a clearer and more organized way, through a scheme, a figure, a better structured table or just described in detail in the manuscript.
- A figure showing the biosynthesis of HBV and the steps in which drugs act by inhibiting the viral cycle would add value to the manuscript.
Reviewer 3 Report
Review of Manuscript # Viruses-1728955
In the manuscript entitled “Direct-acting antiviral agents for Hepatitis C Virus infection-from drug discovery to successful implementation in clinical practice”, the authors summarize the timeline for the discovery of hepatitis C virus (HCV), epidemiology, and basic biology of HCV replication. The authors also list the clinical features of HCV infection, and direct-acting antiviral agents and their working mechanisms. Overall, this well-written paper covers a broad aspect of HCV, from basic biology to clinical medicine. And it should be accepted with minor editing.
Minor comment:
- In line 73, “29.000 HCV-related deaths per year” should be “29,000 HCV-related deaths per year”.
- In lines 100-102, the sentence needs to be rephrased: “Therefore, expression 100 plasmids were generated from HCV cDNA clones encoding 980 amino-terminal amino 101 acid residues of the open reading frame.”
- In lines 114-115, the sentence needs to be rephrased: “While the discovery of viral proteases already shed light on potential therapeutic targets the exact mechanisms behind viral RNA synthesis where still hidden.”
Author Response
Please see attachement

Round 2
Reviewer 2 Report
Dear authors,
The manuscript have improved considerably with the inclusion of the new figures, so I recommend it for publication in Viruses.